Subject Areas:
mechanical engineering/energy/materials science

Keywords:
wear, computational fluid dynamics–discrete element method, buffer effect, large particle, solid–liquid

Author for correspondence:
Zhe Lin
e-mail: linzhe0122@zstu.edu.cn

# Relationship between wear formation and large-particle motion in a pipe bend

Yi Li[1], Hebing Zhang[1], Zhe Lin[1], Zhaohui He[2], Jialiang Xiang[1] and Xianghui Su[1]

[1]The Province Key Laboratory of Fluid Transmission Technology, Zhejiang Sci-Tech University, Hangzhou 310018, People's Republic of China
[2]Zhejiang Institute of Mechanical and Electrical Engineering Co. Ltd, Hangzhou 310000, People's Republic of China

 YL, 0000-0002-7690-6621; ZL, 0000-0002-7422-7799

Fine and large particles flowing through a bend in a pipe move differently and therefore erode the pipe differently. This paper simulates solid–liquid two-phase flow containing large particles in a bend and analyses the relationship between the wear formation and particle motion. Wear experiments are carried out using 3-mm glass bead particles at a mass concentration of 1–15%. At the same time, the flow field and the motion of the granular system are obtained in computational fluid dynamics–discrete element method simulation. The wear formation mechanism is revealed by comparing experiments with numerical simulations. The wear rate of the wall surface increases with the mass concentration, while the marginal growth rate decreases as the mass concentration increases. As the mass concentration increases to a certain value, the degree of wear reaches a maximum and remains unchanged subsequently because of the formation of a particle barrier along the bend wall. The particles near the wall region will bounce forward because of the periodic disturbance flow around particles. The impact of mass bouncing particles causes the formation of the erosion ripple on the test sheet.

## 1. Introduction

Ore pipeline transportation is an inexpensive and highly efficient delivery method in large-scale mining operations. As of 2017, there were about 10 000 km of ore pipelines around the world [1]. Wearing of the pipe wall is unavoidable in the process of ore transportation and has been well studied over the past few decades.

Pipe wear has been experimentally investigated. Kesana *et al.* [2] measured the sand distribution in the bend by sampling

with a probe and analysed the effect of concentration on wall erosion and particle size for particle sizes of 150 µm and 300 µm. Mayank Patel [3] used 50 µm alumina particles in blast erosion experiments on boiler-tube steel, finding that alumina has a higher erosion rate at an impact angle of $30°$ than at an impact angle of $90°$. Asgharpour et al. [4] studied the wear of two series bend in a gas–solid–liquid three-phase flow using particles with a diameter of 300 µm and observed the maximum erosion in the first bend was more than that in the second bend, and the erosion patterns in the two bends are slightly different. Zhang et al. [5] investigated the erosion wear mechanism of high-pressure pipes based on its macroscopic features and scanning electron microscope (SEM) at different positions of the pipes for particle diameters of 300–425 µm and found the effects of impact angle and particle velocity on the wear. Araoye [6] investigated the effects of flow parameters and particle size in solid–liquid two-phase flow on the wear of carbon-steel tubes in experiments and numerical simulations for a particle size of 50–400 µm. Vieira et al. [7] performed ultrasonic measurements to detect the wear of a pipe bend in the case of a gas–liquid–solid three-phase mixture for particle sizes of 20, 150 and 300 µm. Zouaoui et al. [8] found the relationship between the pressure gradient forces and the mixture velocity was substantially different from that of pure-liquid flow in an experiment for particle sizes of 5, 6, 10 and 15 mm.

Numerical simulation has been conducted for two-phase wear. Coker [9] considered the effect of collisions between particles on the wear of horizontal pipes for particle diameters of 50–300 µm. Lin [10,11] used a two-way URANS (SST)-DPM method to investigate gas–solid flow properties and erosion characteristics in cavities.

Zhu et al. [12] constructed a discrete phase model (DPM) to simulate bends due to a gas–liquid two-phase mixture flowing in a bend for particle diameters of 0.5, 0.75 and 1 mm. Peng et al. [13] used a two-way coupled Euler–Lagrange method to solve the problem of liquid–solid flow through a bend and compared wear patterns obtained in experiments for a particle diameter of 450 µm. They illustrated the relationship between the Stokes number and the dynamic motion of the maximum erosion position and proposed three collision mechanisms that explain how the change in the Stokes number affects the erosion position. Zhang et al. [14] examined the erosion of bends due to the flow of a gas–liquid–solid mixture in numerical simulation, analysing the effects of the Stokes number on the particle trajectory and the erosion scar. The mixture contained 20% water, while particle diameters varied from 0.01 to 0.05 mm.

New methods have been proposed to predict two-phase wear more accurately and quickly. Arabnejad et al. [15] explored the role that gas–solid two-phase flow plays for different target materials. Accounting for the shape and size of particles, they proposed a semi-mechanical wear model for the prediction of the wear of metallic materials. The diameters of the particles used in their experiment were 150 and 300 µm. Uzi et al. [16] developed an ODEM model and conducted numerical simulation to capture the wear of a bend in gas–solid delivery for a particle size of 250 µm. Compared with the three-dimensional computational fluid dynamics (CFD)–discrete element method (DEM) model, the ODEM model allows much faster computation. Florio et al. [17] built a computational fluid dynamic model based on compressible gas–solid particles of two-phase flow by combining the rolling, twisting, sliding and adhesion effects of particle interactions to investigate particle-to-wall collisions for a particle diameter of 300 µm. Wang et al. [18] established a new mathematical model to simulate the abrasion of pipe bends transporting solid–liquid two-phase flow in oil pipelines. The particle diameter was 150 µm and an erosion model was proposed based on the Stokes number. Beinert et al. [19] conducted extended contact analysis considering impacts, torsion, shearing and rolling for particle diameters of 0.8 and 1.2 mm.

In summary, most recent studies have focused on gas–solid flow and solid–liquid mixture flow for particle sizes less than 1 mm. There has been little research on the wearing of a pipe wall when a mixture contains large particles. Jankovic [20] investigated particles with a diameter of 5 mm, while Ojala et al. [1] analysed the wear resistance of different commodity steels in the case of 10-mm particles. Nguyen et al. [21] pointed out that the wear mechanism and wear rate of particles depend on the particle size; specifically, when the particle size is within 50–700 µm, the wear surface changes from a W to U shape with an increase in particle size. It is therefore inappropriate to apply the results obtained in research on the gas–solid flow and solid–liquid flow of fine particles to research on large particles. It is necessary and meaningful to study the wear law and motion rule of solid–liquid two-phase flow containing large particles in a bend.

This paper investigates the effect of a solid–liquid mixture containing large particles with a diameter of 3 mm on the wearing of a wall by conducting wear experiments and CFD-DEM numerical simulation.

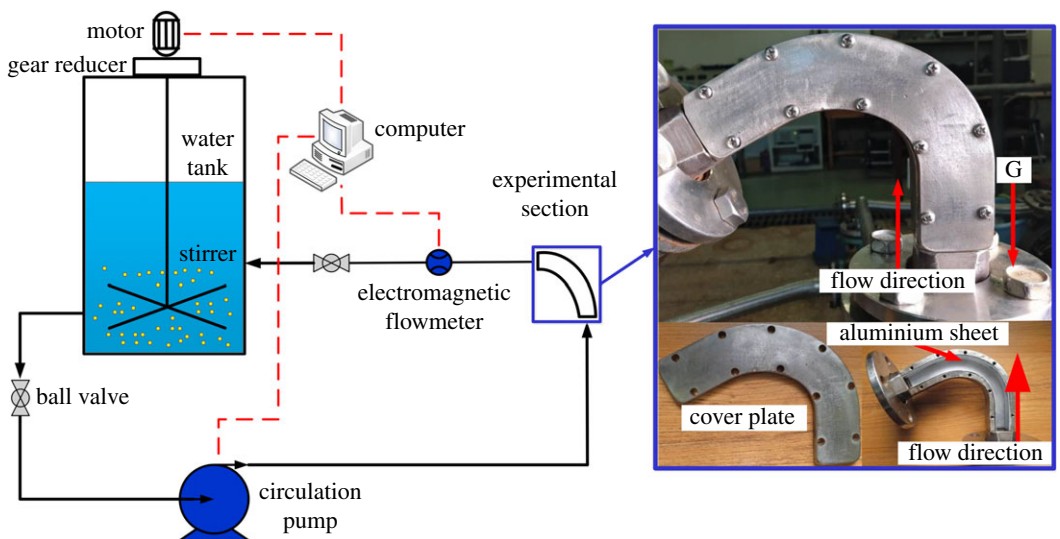

**Figure 1.** Test facility.

## 2. Experimental device and scheme

The experimental loop is shown in figure 1. The experimental section is designed to be removable on one side and fastened with screws. To observe the wear situation, the experimental section is designed as a detachable square pipe. An aluminium sheet is fixed on the wall as a test sample by a groove in the bend. The aluminium sheets have a surface roughness of Ra3.2 and are supplied by a local supplier. All aluminium sheets are processed in a uniform procedure before and after the experiment: (1) rinse the surface of the aluminium sheet with water; (2) gently wipe the surface with a soft cloth soaked with water-diluted detergent; (3) rinse the surface with plenty of water to wash away the dirt; (4) dry the aluminium sheet. The hardness of aluminium is low and the wear resistance is poor. After the experiment, the obvious wear morphology can be seen on the surface of the aluminium sheet. The cross-sectional dimensions of the bend are 22 mm × 22 mm. The bend is oriented vertically in the experiment. There are 600-mm square tube sections before and after the test section. Before the start of the experiment, we inject 2-m$^3$ water into the water tank, then put the experimental particles into the water tank and finally turn on the stirrer to stir. After the stirring is even and stable, we open the two ball valves, then turn on the circulation pump to start the whole test system.

Figure 2$a$ shows the spherical particles used in the experiment. The material is soda-lime glass with a density of 2580 kg m$^{-3}$. Measurements of the diameters of 100 particles varied between 2.80 and 3.15 mm. Figure 2$b$ shows that the fitting curve obeys a normal distribution with a mathematical expectation $\mu$ of 3.00228 and a variance $\sigma^2$ of 0.11045, which meet experimental requirements. The ratio of the diameter of the particle to the diameter of a corresponding circular pipe is 1:8.3 if the square tube is converted to a round pipe with equal area. Fifteen groups of wear experiments are carried out with the mass concentration varying from 1% to 15% in steps of 1%. Each group of experiments is carried out five times. The duration of a wear experiment is 25 min and the mixture flow rate is about 11.77 m s$^{-1}$. Test sheets are weighed one by one before the experiment. The accuracy of the electronic scale is 0.01 g.

## 3. Numerical simulation method

### 3.1. Basic assumptions

The following assumptions are made for the numerical simulation of solid–liquid two-phase flow in a curved pipe.

(1) The solid phase and liquid phases are soda-lime glass particles with a diameter of 3 mm and pure water, respectively, and the physical properties of each phase are constant.
(2) All glass particles are spherical and no phase change occurs.

(a)

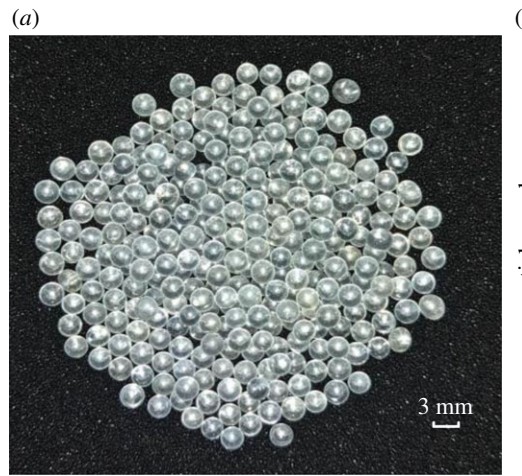

3 mm

(b)

**Figure 2.** (a) Glass particles and (b) particle diameters.

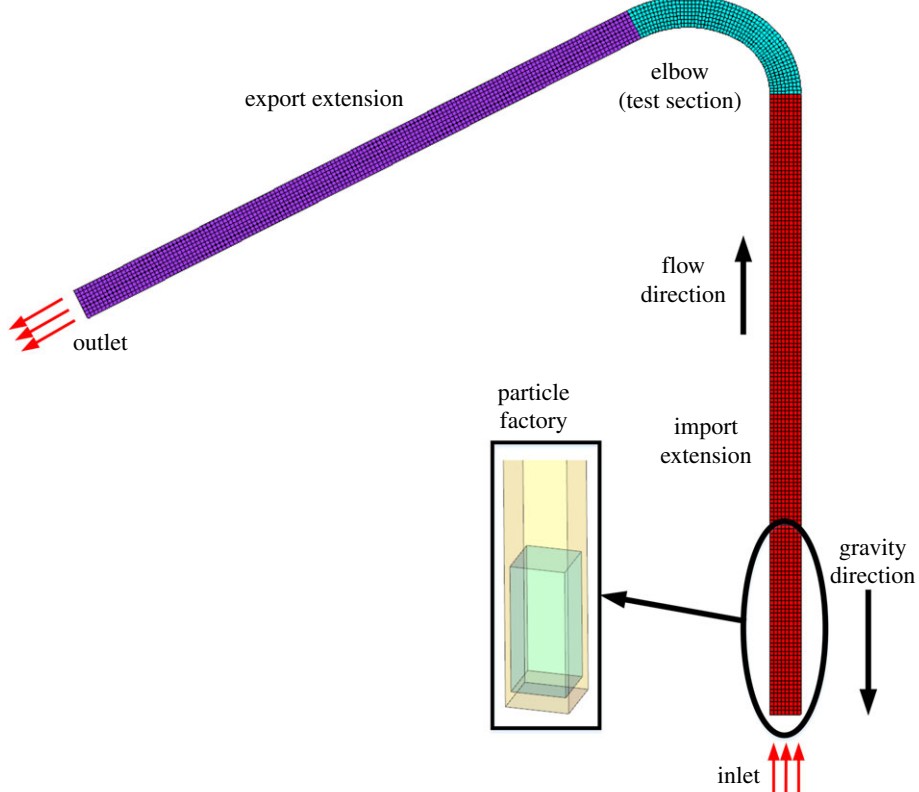

**Figure 3.** Schematic of the geometry and the CFD mesh of the bend used in the simulations.

## 3.2. Models and grids

The variable curvature pipe model is shown in figure 3. To obtain fully developed flow, the inlet and outlet each extend 20 times the length of the section side. The calculation area therefore includes the extended inlet section, the test section and the extended outlet section. In this paper, a structured hexahedral mesh is implemented using integrated computer engineering and manufacturing to ensure high stability and accuracy. Because of the high Reynolds number, the value of $y+$ is set to 30. A progressive mesh is adopted to mesh the boundary with 25-layer grids in the radial direction. The height of the first grid is 0.05 mm and the growth factor is 1.2. To ensure that the number of cells does not affect the results, simulations are conducted with various cell numbers using the average wear amount of the wall surface as the quantitative index. The final number of cells is determined to be 305 166.

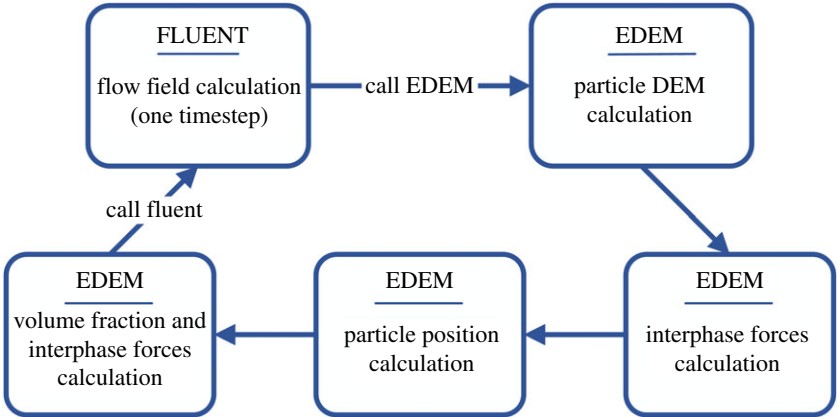

**Figure 4.** EDEM–FLUENT coupling schematic.

## 3.3. Numerical calculation method

The two-phase flow is simulated by coupling the discrete element EDEM particle software and FLUENT software. The water phase is treated as a continuous medium, while particles are regarded as discrete units. The Euler–Lagrange approach is used in our numerical simulation. The flow field is solved using FLUENT and the particle motion is calculated using EDEM. As shown in figure 4, the flow field is calculated by Fluent and the computational result of the fluid is input into the EDEM after it converges. The element EDEM is applied to generate particles and fill the model with particles at a certain rate. The interphase force is used to calculate the particle motion. After a certain number of calculation time steps, the particle calculation result is returned to FLUENT for calculation of the flow field, and the particle motion in the pipe is calculated cyclically.

The two-way coupling method was used in this research. The force of particles applied on the fluid is considered in the two-way coupling method. The motion of the flow fluid can be obtained with the local mean variables according to the continuity and momentum conservation equations. The governing equations of the fluid are given as follows:

Continuity equation:

$$\frac{\mathrm{d}(\alpha_f \rho_f)}{\mathrm{d}t} = 0. \tag{3.1}$$

Momentum conservation equation:

$$\frac{\mathrm{d}(\alpha_f \rho_f \mathbf{u})}{\mathrm{d}t} = -\nabla p + \alpha_f \mu_{\mathrm{eff}} \Delta \mathbf{u} + \alpha_f \rho_f \mathbf{g} + \mathbf{F}_s, \tag{3.2}$$

where $\rho_f$ is the fluid density and is a constant because the fluid is assumed to be incompressible; $u$ is the fluid velocity; $p$ is the pressure of the fluid; $\mu_{\mathrm{eff}}$ is the effective viscosity; $x$ is the coordinate; $\mathbf{g}$ is the acceleration due to gravity; $\mathbf{F}_s$ is the interaction term due to the force between the particles and the fluid; $\alpha_f$ is the porosity near the particle and can be calculated as follows:

$$\alpha_f = 1 - \sum_{i=1}^{n} \frac{V_{p,i}}{V_{\mathrm{cell}}}, \tag{3.3}$$

where $V_{p,i}$ is the volume of particle i in the selected CFD cell; $n$ is the amount of particles inside the cell; $V_{\mathrm{cell}}$ is the volume of the cell.

The translational and rotational motions of the particles are calculated using Newton's kinetic equations:

$$m\frac{\mathrm{d}\mathbf{v}}{\mathrm{d}t} = m\mathbf{g} + \sum \mathbf{F}_c + \mathbf{F}_{\mathrm{drag}} + \mathbf{F}_m + \mathbf{F}_{sl} \tag{3.4}$$

$$\mathbf{I}\frac{\mathrm{d}\omega}{\mathrm{d}t}\sum \mathbf{T}_c + \mathbf{T}_f, \tag{3.5}$$

where $\mathbf{F}_c$, $\mathbf{F}_{\mathrm{drag}}$, $\mathbf{F}_m$ and $\mathbf{F}_{sl}$ are the contact force, fluid drag force, Magnus force and Saffman lift force, respectively. $m$ and $\mathbf{I}$ are the mass and moment of inertia of the particles , respectively. $\mathbf{T}_c$ and $\mathbf{T}_f$ are the contact torque and the torque generated by the fluid phase, respectively.

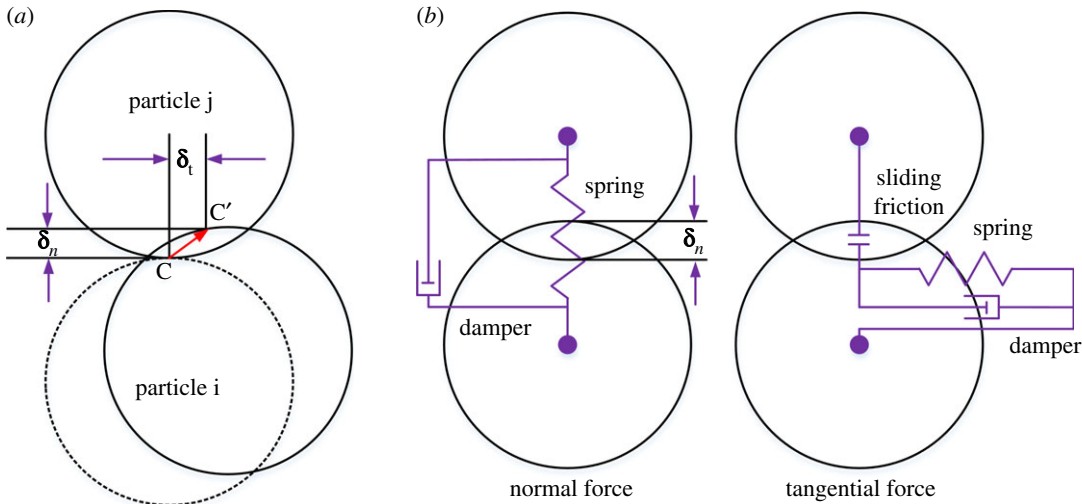

**Figure 5.** (a) Normal displacement $\delta_n$ and tangential displacement $\delta_t$ of particle collision; (b) constitutive model of the interaction between two particles.

Collisions between particles can be illustrated using the soft-sphere model of Zeng [22]. As shown in figure 5a, in a soft-sphere model, a particle is affected by fluid contact with particle j at point C. These two particles are deformed in subsequent movements. Let $\delta_n$ and $\delta_t$ denote the normal displacement and tangential displacement between particles, respectively. $\mathbf{F}_{c,n}$ and $\mathbf{F}_{c,t}$ are the corresponding normal contact force and tangential contact force, respectively. Figure 5b shows the constitutive model of the interaction between the two particles. A spring is used to simulate deformation damp, and a damper is used to simulate the damping effect. When the diameter of one of the particles in the soft ball model is infinite, we can think of the particle as a wall, so that we can simulate the collision between the particle and the wall.

The contact force $\mathbf{F}_c$ can be calculated as follows:

$$\mathbf{F}_c = \mathbf{F}_{c,n} + \mathbf{F}_{c,t}. \tag{3.6}$$

where $\mathbf{F}_{c,n}$ and $\mathbf{F}_{c,t}$ can be solved using the linear spring-buffer model proposed by Cundall & Strack [23]:

$$\mathbf{F}_{c,n} = -k_n\delta_n - \eta_n V_n \tag{3.7}$$

and

$$\mathbf{F}_{c,t} = -k_t\delta_t - \eta_t V_t, \tag{3.8}$$

where $V_n$ and $V_t$ are the normal relative velocity and tangential relative velocity between the particles, respectively; $k_n$ and $k_t$ are the normal and tangential stiffness of the springs, respectively; and $\eta_n$ and $\eta_t$ are the normal and tangential damping coefficients, respectively.

The contact torque $\mathbf{T}_c$ is equal to the torque $\mathbf{T}_t$ generated by the tangential contact force, which is defined as follows:

$$\mathbf{T}_t = \mathbf{r} \times \mathbf{F}_{c,t}, \tag{3.9}$$

where $\mathbf{r}$ is the radius vector from the centre of the circle to the point of contact.

Forces from particles to fluid can be calculated as follows:

$$\mathbf{F}_S = \frac{-\sum_{i=1}^{n}(\mathbf{F}^i_{\text{drag}} + \mathbf{F}^i_{sl} + \mathbf{F}_m)}{V_{\text{cell}}}, \tag{3.10}$$

where $V_{\text{cell}}$ is the volume of the cell and $n$ is the total number of particles in this cell. Actually, the forces of the liquid acting on the particles will react on the liquid from the particles in each computational cell.

The wall wear is calculated using the wear equation proposed by Archard [24] based on contact mechanics between the particle and wall. The Archard wear model is

$$W = Ks\frac{P}{P_m}, \tag{3.11}$$

where $W$ is the wear volume (mm$^3$), $s$ is the sliding distance, $P$ is the applied load (N) and $P_m$ is the hardness of the wall material (N mm$^{-2}$). The ratio between $P$ and $P_m$ is considered the

**Table 1.** Parameters of solid phase in simulation.

| parameters | value |
|---|---|
| material (particles) | soda-lime glass |
| shape (particles) | sphere |
| Poisson's ratio (particles) | 0.25 |
| shear modulus (particles) (GPa) | 1.96 |
| density (particles)(kg m$^{-3}$) | 2500 |
| material (wall) | Al alloy 6061 |
| Poisson's ratio (wall) | 0.3 |
| shear modulus (wall) (GPa) | 26.5 |
| density (wall) (kg m$^{-3}$) | 2700 |
| radius (particles) (mm) | 1.5 |
| initial velocity (particles) (m s$^{-1}$) | 0 |
| time step (s) | $1 \times 10^{-6}$ |
| mass flow (particles) (kg s$^{-1}$) | 0.0569, 0.1138, 0.1707, 0.2276, 0.2845, 0.3414, 0.3983, 0.4552, 0.5121, 0.5690, 0.6259, 0.6828, 0.7397, 0.7966, 0.8535 |

**Table 2.** Parameters of liquid phase in simulation.

| parameters | value |
|---|---|
| material | water |
| viscosity (kg m$^{-1}$ s$^{-1}$) | 0.001003 |
| density (kg m$^{-3}$) | 998.2 |
| initial average velocity (m s$^{-1}$) | 11.77 |
| Reynolds number | $2.577 \times 10^{-5}$ |
| time step (s) | $5 \times 10^{-5}$ |

real contact area. $K$ is the dimensionless wear constant related to the material itself. The wear constant ($K = 3.685 \times 10^{-4}$) is calculated from the data of Prasad [25].

## 3.4. Boundary conditions

In the FLUENT 15.0, a standard K-$\varepsilon$ model is used to analyse the internal flow. The standard wall function is used to deal with the flow near the walls. The no-slip boundary condition is adopted for the boundary of the wall surface. The inlet of the calculation domain is set to the 'velocity-inlet', and the outlet is set to the 'outflow'. Sub-relaxation is chosen for the iterative calculation of the algebraic equation. The convergence precision is $10^{-5}$. The liquid phase enters the inlet of the vertical section of the bend, and a virtual space (which can be named particle factory) is established at the inlet to generate particles in the EDEM 2.7. Specific parameters are given in tables 1 and 2.

# 4. Results and discussion

## 4.1. Relationship between severe wear and particle movement

After each experiment, test pieces are taken from the wall of the bend for cleaning, drying and weighing. Finally, one piece is selected and ranked in ascending order of concentration for each working condition group, as shown in figure 6.

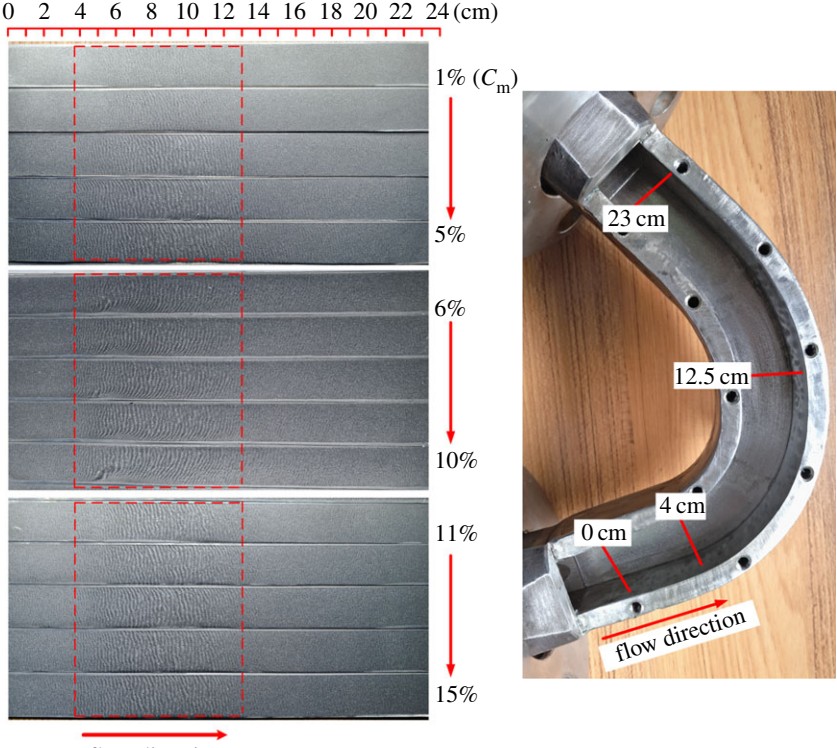

**Figure 6.** Experimental wear results.

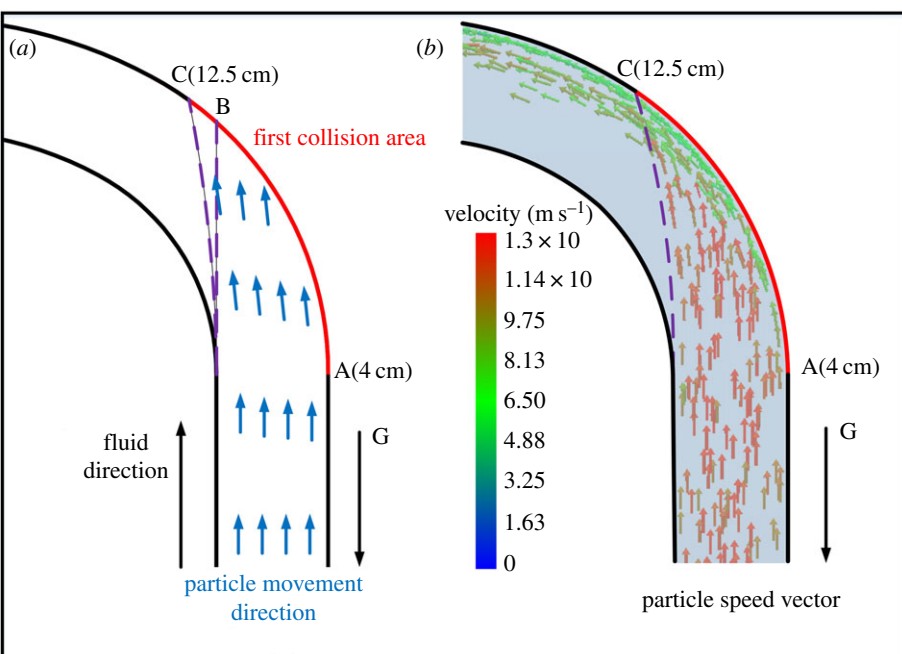

**Figure 7.** (*a*) Schematic diagram of particles moving in the bend and (*b*) particle velocity vector results from CFD-DEM simulation for the condition of 15% mass concentration.

On the surface of each test sheet there is a periodic corrugated pattern that is caused by the collision of particles. The degree of wear varies with working conditions. The severely worn areas mainly exist in the bend section within the interval of 4–12.5 cm, as shown in figure 6. This is also the region where most particles collide with the wall for the first time, as shown in figure 7.

Particles move along the straight pipe section, as shown in figure 7*a*. The first collision with the bend wall occurs on arc AB. However, because of the force of the fluid on particles, the collision area is offset in the flow

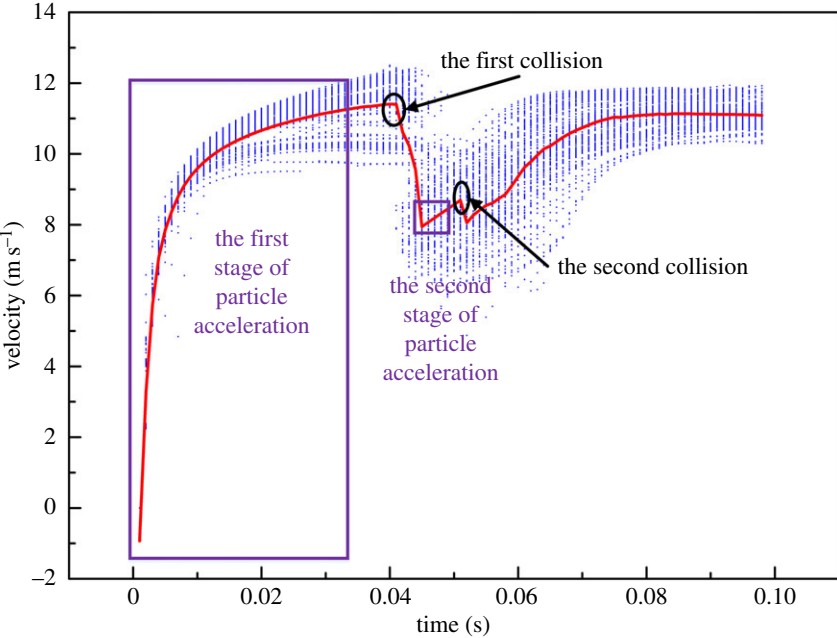

**Figure 8.** Particle velocity magnitude results from CFD-DEM simulation. Scattered blue dots denote discrete velocities, while the red line denotes the mean velocity magnitude in the CFD-DEM simulation.

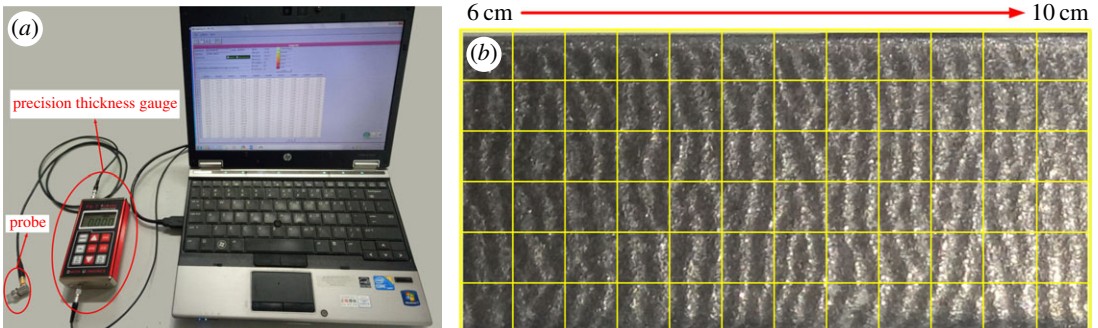

**Figure 9.** (*a*) Precision thickness gauge and (*b*) specific measurement of thickness.

direction by some distance; i.e. arc AC. Figure 7*b* shows particle velocity vectors for a mass concentration of 15%. The figure reveals that the trajectory shifts after particles enter the bend. Particle collisions therefore focus on the area of arc AC, which is the section 4–12.5 cm along the aluminium piece.

Meanwhile, the arc AC is also the region where there is the greatest loss of particle velocity and energy. Wear in this region is therefore most serious. From inlet to outlet of the bend, 130 particles are selected randomly after the flow has stabilized from CFD-DEM simulation. The average velocity of 130 samples is calculated to eliminate accidental errors. The average velocity magnitude of these particles is calculated to illustrate the movement state of the particles in the bend. As shown in figure 8, under the effect of gravity and the fluid drag force, particles move a certain distance in the pipe at higher speed. The particle velocity reduces suddenly when particles collide with the wall surface at this speed for the first time. The wall at this location is thus the most deformed section of wall and even starts to shed material. The particles then rebound from the wall surface at a lower speed and accelerate again under the effect of the drag force of the fluid until colliding a second time with the wall. However, because the acceleration process is short, the velocity is still low when particles collide with the wall for the second time. The first collision therefore leads to the greatest wear.

## 4.2. Relationship between the wear rate and particle mass concentration

To obtain the maximum thickness loss rate of the test sheet, the thickness of the sheet is measured with a PX-7 DL Ultrasonic Thickness Gauge in the region of 6 to 10 cm in the most-worn area. The ultrasonic thickness gauge system is shown in figure 9*a* and the specific measurement method is shown in

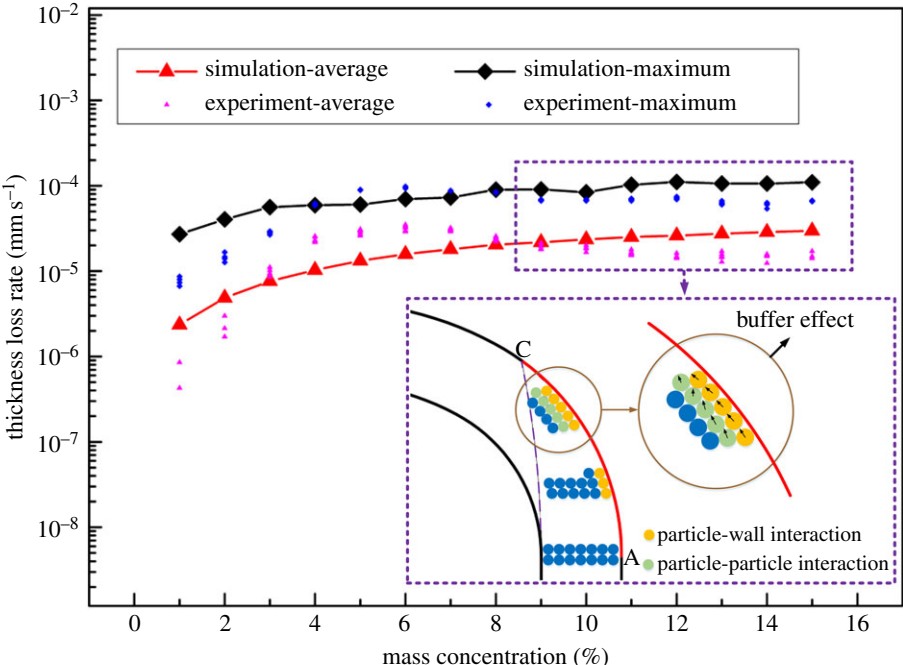

**Figure 10.** Erosion rates at different mass concentrations.

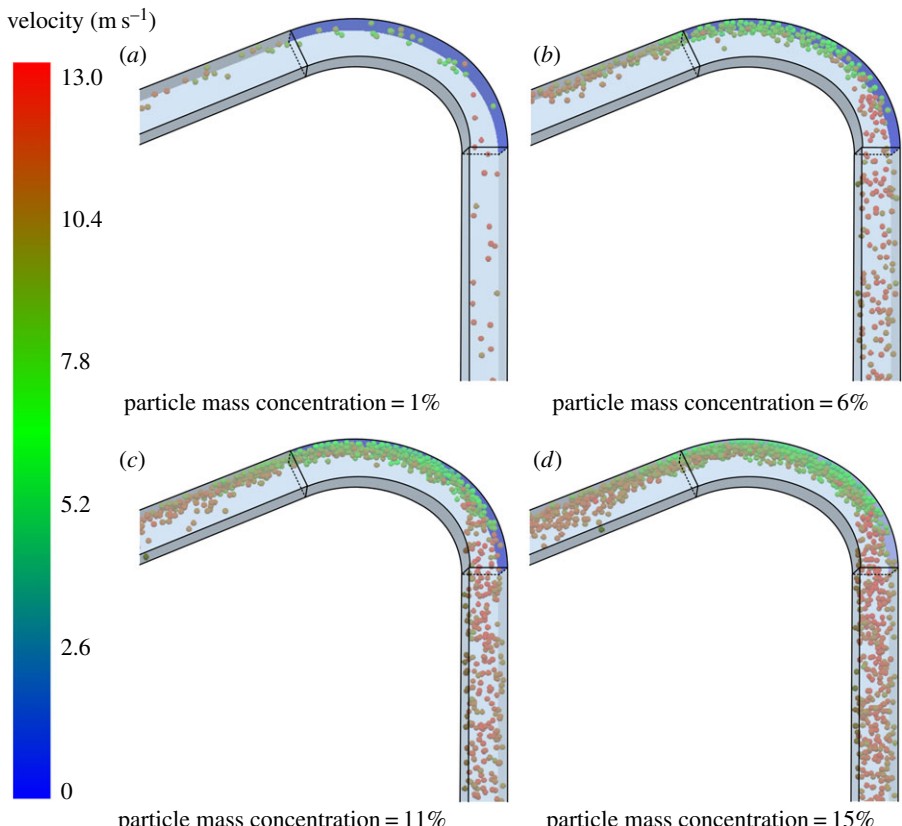

**Figure 11.** Particle motions and positions at different mass concentration results from CFD.

figure 9b. The specimen is divided into 6 and 12 equal parts along the flow direction and the vertical flow direction, respectively. The probe of the ultrasonic thickness gauge is placed at the intersection of the horizontal and vertical lines to measure the thickness of the test sheet. The value of maximum thickness loss is obtained by comparing the minimum material thickness with the original value. The maximum thickness loss rate can then be obtained by dividing the maximum thickness loss by the wear time.

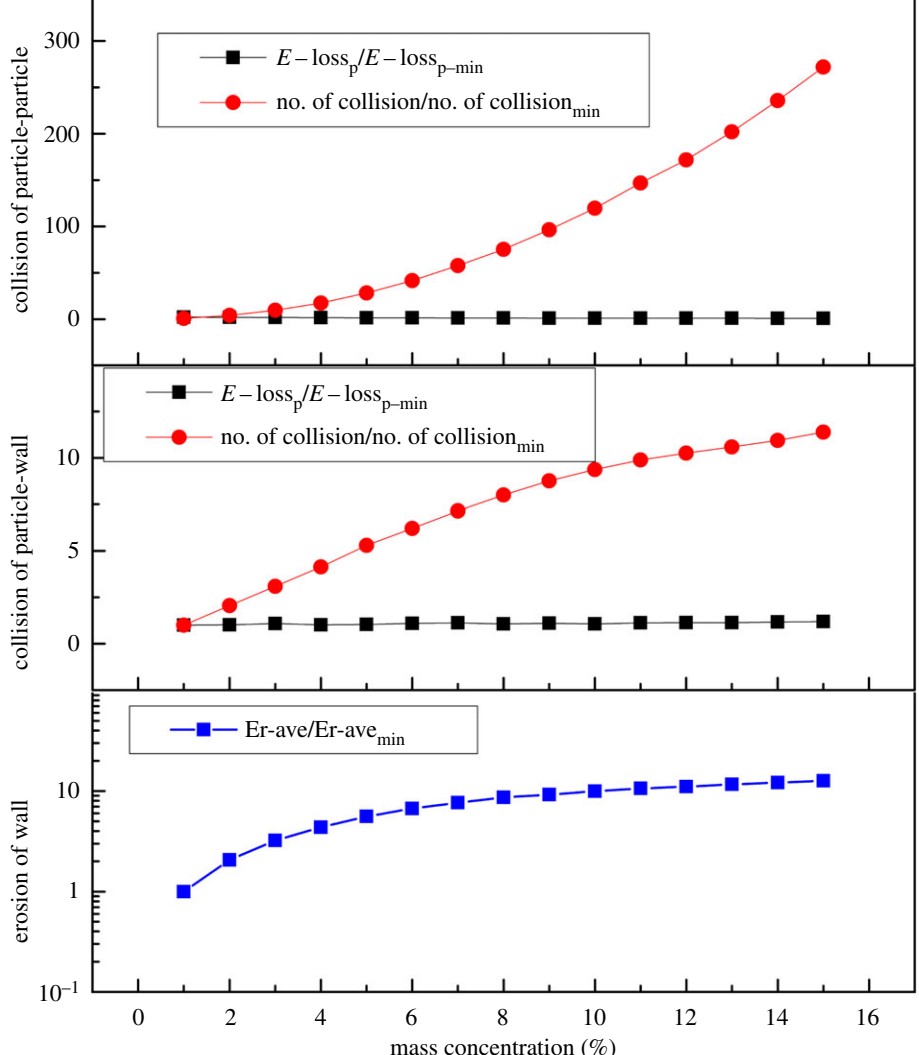

**Figure 12.** Energy loss per particle, number of particles and erosion rate of particle – particle and particle – wall for different mass concentrations. $E - loss_p/E - loss_{p-min}$: energy loss per particle/energy loss per particle min; no. of collision/no. of collision$_{min}$: number of particle collisions/number of particle collisions$_{min}$; Er-ave/Er-ave$_{min}$: average erosion of the wall/average erosion of the wall$_{min}$.

The average thickness loss rate of the test sheet can be obtained by comparing the quality of the sheet before and after the experiment:

$$v = \frac{\Delta m}{\rho st}, \tag{4.1}$$

where $\Delta m$ is the mass of material loss, $\rho$ is the density of the material, $s$ is the surface area of the material and $t$ is time during the wear experiment.

Figure 10 shows the contrast of the average thickness loss rate with the maximum thickness loss rate of the test sheet for different mass concentrations. Results are presented on a logarithmic scale. Figure 11 shows the distribution of particles in the bend section when the particle mass concentration is 1, 6, 11 and 15%.

At each concentration, the maximum erosion rate obtained by the five groups of experiments was basically the same. But the average erosion amount fluctuated, especially under low concentration conditions. The numerical simulation data show that the maximum wear rate and the average wear rate are the same as the concentration changes. A comparison of experimental and numerical simulations of the thickness loss rate in figure 10 shows that the experimental and numerical simulation results are generally consistent. The maximum thickness loss rate under different working conditions is approximately 7–10 times the average thickness loss rate. As the mass concentration of the particles increases gradually from 1% to 9%, the wear rate of the test sheet increases. However,

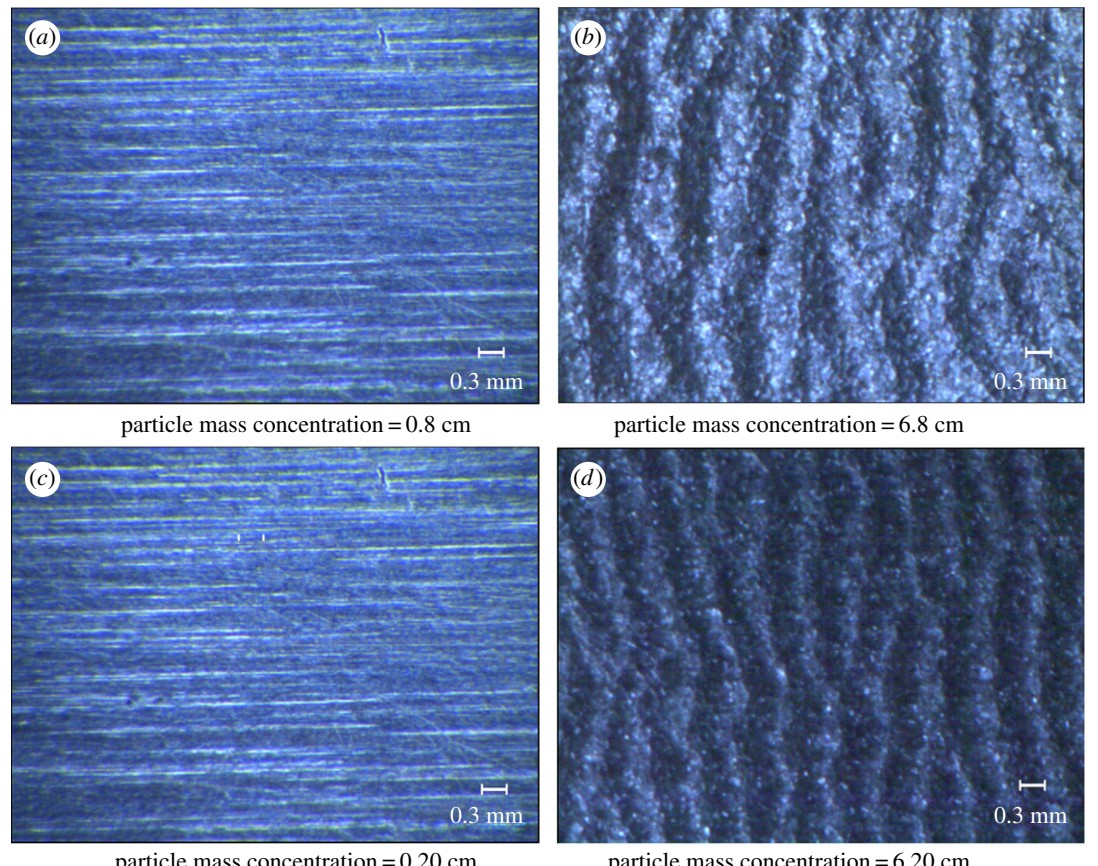

(a) particle mass concentration = 0.8 cm

(b) particle mass concentration = 6.8 cm

(c) particle mass concentration = 0.20 cm

(d) particle mass concentration = 6.20 cm

**Figure 13.** Surface topography of the central position of the test sheet at different distances.

the growth rate gradually decreases with increasing mass concentration. When the mass concentration is greater than 9%, the magnitude of the wear rate tends to be stable, approximately $1 \times 10^{-5}$ mm s$^{-1}$.

This phenomenon can be explained by the buffer effect proposed by Duarte *et al*. [26]. The mechanism of the buffer effect shown in figure 10 is as follows. Firstly, the particles are carried by the fluid into the bend. In this region, the particles move in a straight line, and the collision occurs mainly between particle and particle. Secondly, because the particles are very inertial, their motion is decoupled from the fluid. Therefore, these particles tend to collide with the bend wall directly. After colliding with the wall, the particles are forced to change direction. In this step, the interaction between particles becomes important. Finally, the bend wall accumulates a lot of particles, and the particles adjacent to the wall form a 'virtual barrier' which prevents the surrounding particles from hitting the bend wall directly. The buffering effect occurs at this stage.

As the solid–liquid mixture flows in the pipeline, the factors affecting particle movement are mainly the drag force of the liquid phase, collisions between particles and the wall, and collisions among particles. Figure 11 shows that the particle velocity follows the same law of change for different mass concentrations. This means that the drag force of the liquid phase has the same effect on particles at the same water velocity. The particle velocity is thus almost the same value of about 10 m s$^{-1}$ before particles enter the bend region. Therefore, the change in the wear law is mainly related to the number of collisions of particles and the energy loss (i.e. the deformation energy of the wall surface transmitted by particles). Because of the particles' energy loss during the collision process, the particle velocity decreases and the particles thus gather near the wall surface and travel forward slowly along the surface. Subsequent particles therefore first collide with previous particles, and a large number of particles decelerate to form a barrier near the wall surface, which provides a buffering effect for particle–particle–wall collisions. The barrier becomes more and more obvious as the concentration increases.

In figure 12, the kinetic energy lost due to the collision between particles and particles is almost the same for different concentrations; meanwhile, the average deformation energy of the wall surface obtained from the collision of a single particle with the wall surface is also almost the same for different concentrations. It is concluded that the wear quantity of the wall surface is directly related to

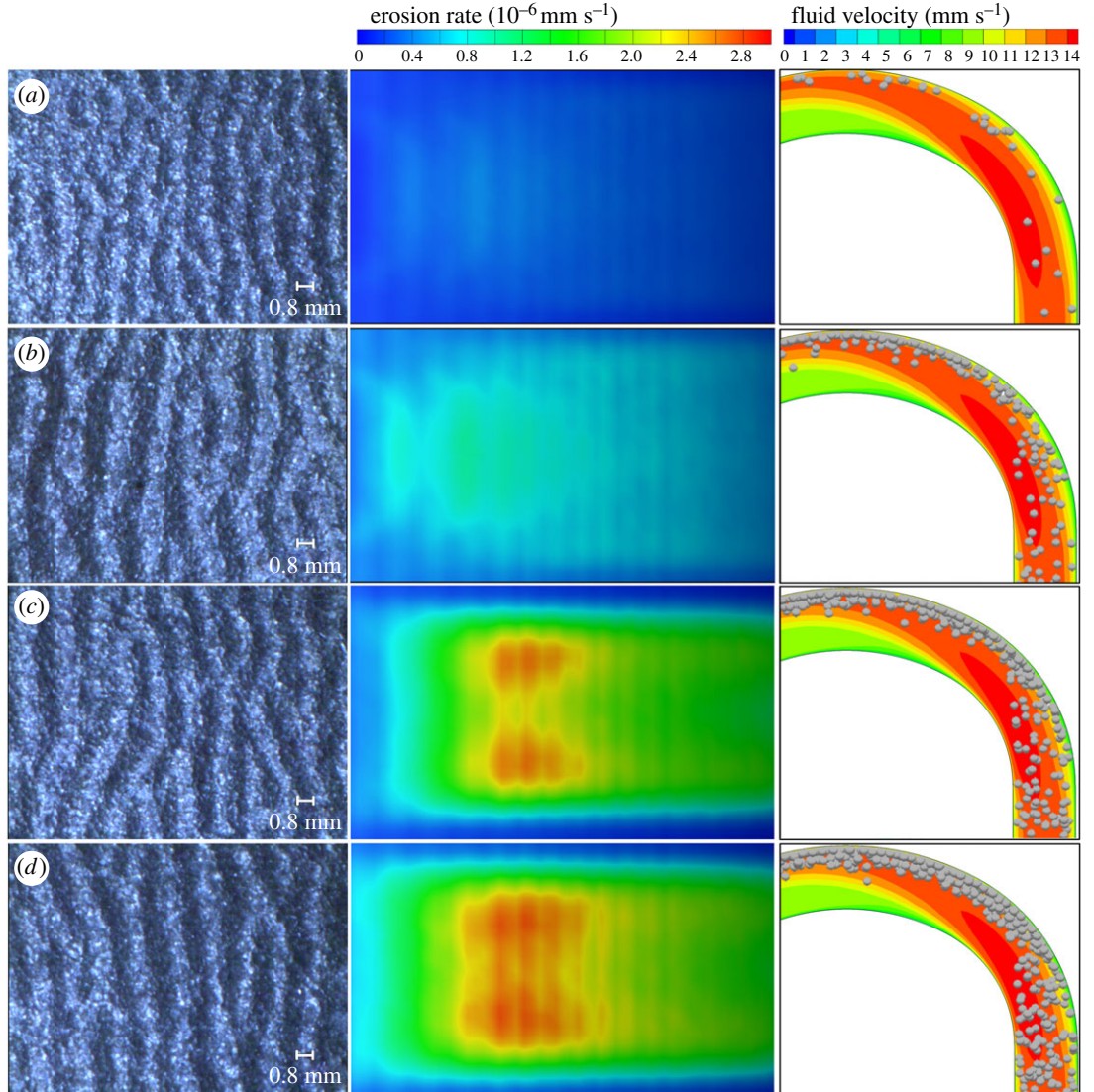

**Figure 14.** Surface topography (experiment), wear cloud (simulation) and particle position in the flow field (simulation) at 8 cm for mass concentrations of (*a*) 1%, (*b*) 6%, (*c*) 11% and (*d*) 15%.

the collision number. The number of collisions between the particles and wall increases with increasing concentration when the concentration is low. However, when the concentration increases to a certain level, the number of collisions between particles and particles increases sharply, forming a barrier layer near the wall surface, so that the collision between particles and the wall tends to be stable. The wear quantity thus tends to stabilize.

## 4.3. Relationship between the surface morphology and solid–liquid mixture motion

To explore the formation mechanism of the wavy streak on the worn wall surface seen in figure 6, a Japanese Keith VHX-2000 super-depth microscope is applied to observe the microscopic appearance of the wear of the surface material.

Figure 13 shows the surface topography at different distances from the entrance. The distance of 8 cm is almost at the beginning of the curve, while the distance of 20 cm is near the outlet of the bend. The wear profile image shows a nearly periodic waveform along the flow direction. The amplitude of corrugation does not change greatly in the same area of worn surface. A comparison of figure 13*b*,*d* shows that the surface topography is similar at different distances. However, we take the total width of the five waves and find that the widths of the two figures are 3.81 and 3.05 mm. The ripple frequency is higher in the latter figure.

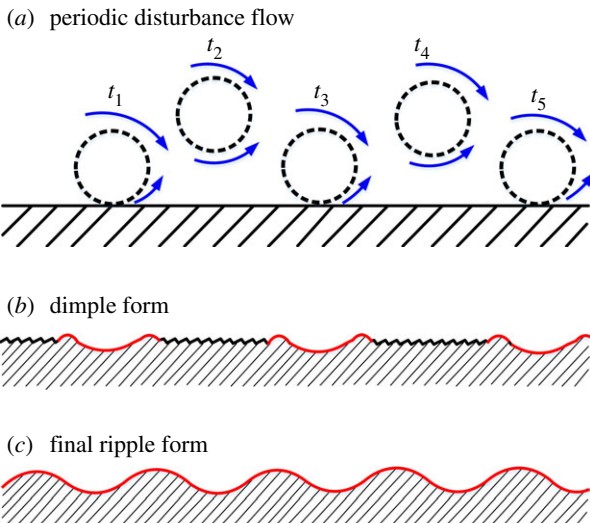

(*a*) periodic disturbance flow

(*b*) dimple form

(*c*) final ripple form

**Figure 15.** Schematic diagram of ripple formation.

Figure 14 shows that the wearing of the test sheet surface has the same pattern for all working conditions, but the degree of wear differs. The experimental and simulation results are consistent. When the mass concentration is low, the amplitude of the wear fringe is low. The fringe amplitude increases as the concentration increases. When the concentration reaches a certain level, the amplitude of the fringe is basically the same, which is consistent with the results presented in the wear cloud diagram.

The erosion ripple is caused by the disturbance of flow around particles according to Karimi & Schmid [27]. Owing to the friction of wall–liquid interface, the velocity of fluid near the wall surface is much slower than that of the bulk fluid. Meanwhile, the particle concentration near the wall is larger than that in the main flow. With the coupling of these two effects, the velocity difference increases and gives rise to shear forces, and thus creates the periodic disturbance of flow around particles (figure 15*a*). Under the effect of flow turbulence, particles will bounce forward (i.e. impact on the wall surface, rebound and then impact again), and leave some dimples on it (figure 15*b*). Over a period of time, there is huge impact of bouncing particles on the surface, thus causing the formation of the erosion ripple (figure 15*c*). The waviness of the erosion ripple is dominated by the particle concentration. The higher the particle concentration, the larger the surface waviness.

# 5. Conclusion

The main findings of the study are as follows:

(1) The region of most serious wear is located near the inlet of the bend where the first collision occurs. At this time, particles enter the bend with the highest velocity, and the energy loss is the largest during the collision process. The velocity of particles drops rapidly afterwards. The subsequent number of collisions and degree of wear are lower.
(2) The degree of wall surface wear increases as the mass concentration increases. However, the marginal growth rate is slower. When the mass concentration increases to a certain value, the degree of wear reaches a maximum and remains unchanged subsequently because of the buffering effect of particles near the wall surface.
(3) Caused by the periodic disturbance flow around particles, particles near the wall region will bounce forward. The impact of mass bouncing particles makes the formation of the erosion ripple on the test sheet. A higher particle concentration leads to a larger waviness.

Data accessibility. Data are available from the Dryad Digital Repository: http://dx.doi.org/10.5061/dryad.j1v7g05 [28].
Authors' contributions. L.Y. carried out the experimental work, numerical simulation, data analysis and drafted the manuscript. Z.H. participated in partial data analysis and drafted the manuscript. L.Z. participated in the numerical simulation and drafted the manuscript. H.Z. conceived of, designed and coordinated the study, and helped draft the manuscript. X.J. and S.X. participated in the design of the study and experimental work. All the authors gave their final approval for publication.

Competing interests. We declare we have no competing interests.

Funding. This research was supported by National Natural Science Foundation of China (grant no. 51576179) and Zhejiang Provincial Natural Science Foundation of China (grant no. LZ15E090002).

Acknowledgements. We thank Lv Wenshuai and Zeng Xiaodong for their help in the experiment and data analysis. We are also grateful to Dr Oliver Jensen, Prof R. Kerry Rowe and two anonymous reviewers, who provided comments that substantially improved the manuscript.

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
