## [Reviewer comments · Royal Society Open Science]

Review History

RSOS-181254.R0 (Original submission)

Review form: Reviewer 1 (Goodarz Ahmadi)

Is the manuscript scientifically sound in its present form?

No

Are the interpretations and conclusions justified by the results?

No

Is the language acceptable?

Yes

Is it clear how to access all supporting data?

No

Do you have any ethical concerns with this paper?

No

Have you any concerns about statistical analyses in this paper?

Yes

Recommendation?

Major revision is needed (please make suggestions in comments)

Comments to the Author(s)

Please see the enclosed file (Appendix A).

Review form: Reviewer 2

Is the manuscript scientifically sound in its present form?

Yes

Are the interpretations and conclusions justified by the results?

Yes

Is the language acceptable?

Yes

Is it clear how to access all supporting data?

Yes

Do you have any ethical concerns with this paper?

No

Have you any concerns about statistical analyses in this paper?

No

Recommendation?

Major revision is needed (please make suggestions in comments)

Comments to the Author(s)

1. In equation 4.7, what was the value of K for the computation? Why?
2. The software versions of EDEM and FLUENT should be mentioned in the manuscript.
3. The computational model was unsteady and the erosion tests were dependent upon time, so all the results should depend upon time from Fig. 10 to Fig. 14. For example in Fig. 10, it is impossible that the erosion rates keep constant all the time.

Decision letter (RSOS-181254.R0)

20-Sep-2018

Dear Dr Li,

The editors assigned to your paper ("Relationship between Wear Formation and Large-particle Motion in an Elbow") have now received comments from reviewers. We would like you to revise your paper in accordance with the referee and Associate Editor suggestions which can be found below (not including confidential reports to the Editor). Please note this decision does not guarantee eventual acceptance.

Please submit a copy of your revised paper before 13-Oct-2018. Please note that the revision deadline will expire at 00.00am on this date. If we do not hear from you within this time then it will be assumed that the paper has been withdrawn. In exceptional circumstances, extensions may be possible if agreed with the Editorial Office in advance. We do not allow multiple rounds of revision so we urge you to make every effort to fully address all of the comments at this stage. If deemed necessary by the Editors, your manuscript will be sent back to one or more of the original reviewers for assessment. If the original reviewers are not available, we may invite new reviewers.

- Data accessibility

If you wish to submit your supporting data or code to Dryad (<http://datadryad.org/>), or modify your current submission to dryad, please use the following link:
<http://datadryad.org/submit?journalID=RSOS&manu=RSOS-181254>

- **Competing interests**

- **Authors' contributions**

- **Acknowledgements**

- **Funding statement**

Please note that Royal Society Open Science charge article processing charges for all new submissions that are accepted for publication. Charges will also apply to papers transferred to Royal Society Open Science from other Royal Society Publishing journals, as well as papers submitted as part of our collaboration with the Royal Society of Chemistry (<http://rsos.royalsocietypublishing.org/chemistry>). If your manuscript is newly submitted and subsequently accepted for publication, you will be asked to pay the article processing charge, unless you request a waiver and this is approved by Royal Society Publishing. You can find out more about the charges at <http://rsos.royalsocietypublishing.org/page/charges>. Should you have any queries, please contact openscience@royalsociety.org.

Kind regards,

Royal Society Open Science Editorial Office
Royal Society Open Science
openscience@royalsociety.org

on behalf of Dr Oliver Jensen (Associate Editor) and Prof. R. Kerry Rowe (Subject Editor)

Associate Editor's comments (Dr Oliver Jensen):

Associate Editor: 1

Comments to the Author:

Please revise your manuscript, paying close attention to the points raised by the reviewers. They have some legitimate concerns over the clarity of the presentation of your study and the validity of your results. Please provide further justification in particular for the results presented in Figure 14. Your revised manuscript will be subject to further review.

Some specific suggestions regarding presentation are as follows. In the title of your manuscript, replace "elbow" with "pipe bend" to avoid confusion with biomechanical applications. There is some erroneous text at the end of section 2, and obvious errors in citations to references on pages 6 and 11.

Comments to Author:

Reviewers' Comments to Author:

Reviewer: 1

Comments to the Author(s)

Please see the enclosed file

Reviewer: 2

Comments to the Author(s)

1. In equation 4.7, what was the value of K for the computation? Why?
2. The software versions of EDEM and FLUENT should be mentioned in the manuscript.
3. The computational model was unsteady and the erosion tests were dependent upon time, so all the results should depend upon time from Fig. 10 to Fig. 14. For example in Fig. 10, it is impossible that the erosion rates keep constant all the time.

Author's Response to Decision Letter for (RSOS-181254.R0)

See Appendices B - D.

RSOS-181254.R1 (Revision)

Review form: Reviewer 1 (Goodarz Ahmadi)

Is the manuscript scientifically sound in its present form?

Yes

Are the interpretations and conclusions justified by the results?

No

Is the language acceptable?

Yes

Is it clear how to access all supporting data?

Not Applicable

Do you have any ethical concerns with this paper?

No

Have you any concerns about statistical analyses in this paper?

No

Recommendation?

Accept with minor revision (please list in comments)

Comments to the Author(s)

Please see the enclosed file (Appendix E).

Review form: Reviewer 2

Is the manuscript scientifically sound in its present form?

Yes

Are the interpretations and conclusions justified by the results?

Yes

Is the language acceptable?

Yes

Is it clear how to access all supporting data?

Yes

Do you have any ethical concerns with this paper?

No

Have you any concerns about statistical analyses in this paper?

No

Recommendation?

Accept with minor revision (please list in comments)

Comments to the Author(s)

It is suggested to simply add the explanation of average erosion rate for each group by mass difference before and after the experiment.

It is also suggested to simply add the explanation of all the computational erosion data was selected after the erosion was stabilized.

Decision letter (RSOS-181254.R1)

13-Nov-2018

Dear Dr Li:

On behalf of the Editors, I am pleased to inform you that your Manuscript RSOS-181254.R1 entitled "Relationship between Wear Formation and Large-particle Motion in a Pipe Bend" has been accepted for publication in Royal Society Open Science subject to minor revision in accordance with the referee suggestions. Please find the referees' comments at the end of this email.

The reviewers and Subject Editor have recommended publication, but also suggest some minor revisions to your manuscript. Therefore, I invite you to respond to the comments and revise your manuscript.

- Ethics statement

- Data accessibility

If you wish to submit your supporting data or code to Dryad (<http://datadryad.org/>), or modify your current submission to dryad, please use the following link:
<http://datadryad.org/submit?journalID=RSOS&manu=RSOS-181254.R1>

- Competing interests

- Authors' contributions

- Acknowledgements

- Funding statement

Because the schedule for publication is very tight, it is a condition of publication that you submit the revised version of your manuscript before 22-Nov-2018. Please note that the revision deadline will expire at 00.00am on this date. If you do not think you will be able to meet this date please let me know immediately.

Supplementary files will be published alongside the paper on the journal website and posted on

the online figshare repository (<https://figshare.com>). The heading and legend provided for each supplementary file during the submission process will be used to create the figshare page, so please ensure these are accurate and informative so that your files can be found in searches. Files on figshare will be made available approximately one week before the accompanying article so that the supplementary material can be attributed a unique DOI.

Please note that Royal Society Open Science charge article processing charges for all new submissions that are accepted for publication. Charges will also apply to papers transferred to Royal Society Open Science from other Royal Society Publishing journals, as well as papers submitted as part of our collaboration with the Royal Society of Chemistry (<http://rsos.royalsocietypublishing.org/chemistry>). If your manuscript is newly submitted and subsequently accepted for publication, you will be asked to pay the article processing charge, unless you request a waiver and this is approved by Royal Society Publishing. You can find out more about the charges at <http://rsos.royalsocietypublishing.org/page/charges>. Should you have any queries, please contact openscience@royalsociety.org.

on behalf of Dr Oliver Jensen (Associate Editor) and Professor R. Kerry Rowe (Subject Editor)
openscience@royalsociety.org

Associate Editor Comments to Author (Dr Oliver Jensen):

Please address all of the remaining points identified by the reviewers in a further revision of your paper.

Reviewer comments to Author:

Reviewer: 1

Comments to the Author(s)

Please see the enclosed file

Reviewer: 2

Comments to the Author(s)

It is suggested to simply add the explanation of average erosion rate for each group by mass difference before and after the experiment.

It is also suggested to simply add the explanation of all the computational erosion data was selected after the erosion was stabilized.

Author's Response to Decision Letter for (RSOS-181254.R1)

See Appendix F.

Decision letter (RSOS-181254.R2)

26-Nov-2018

Dear Dr Li,

I am pleased to inform you that your manuscript entitled "Relationship between Wear Formation and Large-particle Motion in a Pipe Bend" is now accepted for publication in Royal Society Open Science.

on behalf of Dr Oliver Jensen (Associate Editor) and R. Kerry Rowe (Subject Editor)
openscience@royalsociety.org

Follow Royal Society Publishing on Twitter: [@RSocPublishing](https://twitter.com/RSocPublishing)

Appendix A

Comments on Manuscript Number: RSOS-181254

Title: Relationship between Wear Formation and Large particle Motion in an Elbow

This manuscript reports a study of large particle motions in a bend and their effects on wear formation. The authors performed an experimental study as well as using the ANSYS-Fluent and DEM commercial software. While the results are potentially useful there are a number of areas of concern. There are:

- On the numerical simulations part, there is no mention of flow regime. One may infer that the flow is in turbulent regime, as there is one on page 7 that the k-e model was used. There is no discussion on how the interaction of flow turbulence and particles was accounted for. If the turbulence fluctuation effects are ignored, then the validity of the results is questionable.
- The flow and particle boundary conditions are not fully discussed. It was just mentioned on top of page 7 that the no-slip boundary condition was used. It seems that the authors are not familiar with the options of the code they are using for the turbulent flow.
- Interactions of particles with wall also need more explanation.
- It is not quite clear what are shown in Figure 8.
- The wear model used needs to be more fully described. Also the schematic shown in Figure 10 needs to be more clearly described.
- The experimentally observed wavy nature of surface obtained in Figure 6 is interesting, but needs to be justified if these are due to the mechanics of the erosion process or defect of the materials used.
- Figure 14, if justified, is very interesting, but both the experimental and numerical results are suspect. It is quite doubtful if the random particle collisions with the bend will lead to periodic ripples for erosion shown in the figure for turbulent flows.

Appendix B

Written Answer for the Reviewers

Dear Editor

Thank you for your letter. We thank the reviewers for the time and effort that they have put into reviewing the previous version of the manuscript. Their suggestions have enabled us to improve our work. Based on the instructions provided in your letter, we uploaded the file of the revised manuscript. Accordingly, we have uploaded a copy of the original manuscript with part of the changes highlighted in MS Word.

Appended to this letter is our point-by-point response to the comments raised by the reviewers. The comments are reproduced and our responses are given directly afterward in a different color. We hope that the revised manuscript is accepted for publication in the ROYAL SOCIETY OPEN SCIENCE.

Sincerely,

Yi Li, Hebing Zhang, Zhe Lin, Zhaohui He, Jialiang Xiang, Xianghui Su
linzhe0122@zstu.edu.cn

Editor's comments

Thank you for your interesting comments. Your comments are helpful to further improve the manuscript. We carefully reviewed the manuscript and modified it as required.

Associate Editor's comments

1. Please provide further justification in particular for the results presented in Figure 14.

Re: Thank you for your suggestion. We have further described the results in Figure 14. In addition, we reinterpreted schematic diagram of ripple formation. (Page 15.)

2. In the title of your manuscript, replace "elbow" with "pipe bend" to avoid confusion with biomechanical applications.

Re: We replaced "elbow" with "pipe bend" or "bend" in the manuscript as required.

3. There is some erroneous text at the end of section 2.

Re: We found the erroneous text at the end of section 2 and deleted it in the manuscript. (Page3)

4. There is some obvious errors in citations to references on pages 6 and 11.

Re: We carefully reviewed the citations to references on pages 6 (Reference [21]) and 11 (Reference [24]) and modified them.

Appendix C

Written Answer for the Comment 1

Dear Comment 1

Thank you for your letter. We thank you for the time and effort that you have put into reviewing the previous version of the manuscript. Your suggestions have enabled us to improve our work. Based on the instructions provided in your letter, we uploaded the file of the revised manuscript. Accordingly, we have uploaded a copy of the original manuscript with part of the changes highlighted in MS Word.

Appended to this letter is our point-by-point response to the comments. The comments are reproduced and our responses are given directly afterward in a different color. We hope that the revised manuscript is accepted for publication in the ROYAL SOCIETY OPEN SCIENCE.

Sincerely,

Yi Li, Hebing Zhang, Zhe Lin, Zhaohui He, Jialiang Xiang, Xianghui Su
linzhe0122@zstu.edu.cn

Reviewer comments 1

1. On the numerical simulations part, there is no mention of flow regime. One may infer that the flow is in turbulent regime, as there a one on page 7 that the k-e model was used.

Re: Thank you for your comment. Reynolds number is $2.577e5$, we have added it in Table 2 to illustrate the turbulent regime. (Page7)

2. There is no discussion on how the intersection of flow turbulence and particles was accounted for. If the turbulence fluctuation effects are ignored, then the validity of the results is questionable.

Re: Thank you for your suggestion.

1)The two-way coupling method was used in this research.

The force of particles applied on the fluid is considered in the two-way coupling method. The motion of the flow fluid can be obtained with the local mean variables according to the continuity and momentum conservation equations.

The governing equations of the fluid are given as follows:

Continuity equation:

$$\frac{\partial}{\partial t}(\alpha_f \rho_f) + \frac{\partial}{\partial x_j}(\alpha_f \rho_f u_j) = 0$$

Momentum conservation equation:

$$\frac{\partial}{\partial t}(\alpha_f \rho_f u_i) + \frac{\partial}{\partial x_j}(\alpha_f \rho_f u_i u_j) = -\frac{\partial p}{\partial x_i} + \frac{\partial}{\partial x_j} \left[\alpha_f \mu_{eff} \left(\frac{\partial u_i}{\partial x_j} + \frac{\partial u_j}{\partial x_i} \right) \right] + \alpha_f \rho_f \mathbf{g} + \mathbf{F}_s$$

where ρ_f represents the fluid density and is a constant because the fluid is assumed to be incompressible; u represents the fluid velocity; p represents the pressure of the fluid; μ_{eff} represents the effective viscosity; x represents the coordinates; \mathbf{g} is the acceleration due to gravity; \mathbf{F}_s is the interaction term due to the force between the particles and the fluid; α_f is the porosity near the particle and can be calculated as:

$$\alpha_f = 1 - \sum_{i=1}^n V_{p,i} / V_{cell}$$

where $V_{p,i}$ represents the volume of particle i in the selected CFD cell; n represents the amount of particles inside the cell; V_{cell} represents the volume of the cell.

2) Forces and torques from the fluid to the particles

The translational and rotational motions of the particles are calculated using Newton's kinetic equations:

$$m \frac{d\mathbf{v}}{dt} = m\mathbf{g} + \sum \mathbf{F}_c + \mathbf{F}_{drag} + \mathbf{F}_m + \mathbf{F}_{sl}$$

$$\mathbf{I} \frac{d\boldsymbol{\omega}}{dt} = \sum \mathbf{T}_c + \mathbf{T}_f$$

where \mathbf{F}_c , \mathbf{F}_{drag} , \mathbf{F}_m and \mathbf{F}_{sl} respectively represent the contact force, fluid drag force, Magnus force and Saffman lift force. m and \mathbf{I} are respectively the mass and moment of inertia of the particles. \mathbf{T}_c and \mathbf{T}_f respectively denote the contact torque and the torque generated by the fluid phase.

3) Forces from particles to fluid

In the study of this paper, the maximum volume concentration of solid particles is 6.13%. In this case, the force can greatly influence the fluid flow, and the two-way coupling method must be used.

$$\mathbf{F}_s = \frac{-\sum_{i=1}^n (\mathbf{F}_{drag}^i + \mathbf{F}_{sl}^i + \mathbf{F}_m)}{V_{cell}}$$

where V_{cell} is the volume of the cell and n represents the total number of particles in this cell. Actually, this equation is based on the principle of Newton's third law of motion such that the forces of the liquid acting on the particles will react on the liquid from the particles in each computational cell.

3. The flow and particle boundary conditions are not fully discussed. It was just mentioned on top of page 7 that the no-slip boundary condition was used. It seems that the authors are no familiar with the options of the code they are using for the turbulent flow.

Re: We are grateful for the suggestion. Your comment is helpful to further improve the manuscript. We have added in section of 4.4(Page 7). The changes of the manuscript have been highlighted in yellow text.

4. Interactions of particles with wall also need more explanation.

Re: Thanks for your suggestion.

According to the soft ball model, when the particle i collides with the wall surface, the normal force of the wall surface to the particle is:

$$\mathbf{F}_{c,n} = -k_n \delta_n - \eta_n \mathbf{v}_n$$

The normal force acting on a particle by its wall is:

$$\mathbf{F}_{c,t} = -k_t \delta_t - \eta_t \mathbf{v}_t$$

where \mathbf{V}_n and \mathbf{V}_t are the normal relative velocity and tangential relative velocity between the particles; k_n and k_t are respectively the normal and tangential stiffness of the springs; and η_n and η_t are respectively the normal and tangential damping coefficients. δ_n and δ_t respectively denote the normal displacement and tangential displacement between particles.

The reaction force of the particle to the wall surface is equal to the force of the wall surface to the particle. The wall is treated as an infinitely sized particle, so its mass is also infinite. Because of the finite force, the acceleration of infinite particle (wall) is 0. In addition, the initial velocity of the wall is 0, so the wall remains stationary.

5. It is not quite clear what are shown in Figure 8.

Re: Thank you for your comment. We have detailed the data source of Figure 8 and the purpose to be explained. From inlet to outlet of the bend, 130 particles are selected randomly after the flow has stabilized from CFD-DEM simulation. The average velocity magnitude of these particles is calculated to illustrate the movement state of the particles in the bend. (Page 10)

6. The wear model used need to be more fully described. Also the schematic shown in Figure 10 need to be more clearly described.

Re: Thank you for your suggestion .

The wear model has been re-described more fully (Page 7). In addition, the buffer effect shown in the schematic diagram of figure 10 is also described in more detail. (Page 12)

7. The experimentally observed wavy nature of surface obtained in Figure 6 is

interesting, but need to be justified if these are due to the mechanics of the erosion process or defect of the materials used.

Re: We are grateful for the suggestion. We have added the processing of experimental materials in the manuscript. In addition, we have added the appearance of the aluminum sheet before the experiment in Fig.6. The aluminum sheets have a surface roughness of Ra3.2 and are supplied by local suppliers. All aluminum sheets are processed in a uniform procedure before and after the experiment:

- 1). Rinse the surface of the aluminum sheet with water;
- 2). Gently wipe the surface with a soft cloth soaked with water-diluted detergent;
- 3). Rinse the surface with plenty of water to wash away the dirt;
- 4). Dry the aluminum sheets.

So all the aluminum sheets meet the experimental requirements.(Page 3)

8. Figure 14, if justified is very interesting, but both the experimental and numerical results are suspect. It is quite doubtful if the random particle collisions with the bend will lead to periodic ripples for of erosion shown in the figure for turbulent flows.

Re: We are grateful for the suggestion.

1) Experiment: The aluminum sheets have a surface roughness of Ra3.2. All aluminum sheets are processed in a uniform procedure before and after the experiment, to ensure that the aluminum sheet is in accordance with the experimental standards (Page 3). In addition, each group of experiments is carried out five times and the duration of the wear test is the same, to eliminate the random error. (Page 4)

2) Simulation: The two-way coupling method was used in this research. The Euler-Lagrange approach is used in our numerical simulation. The flow field is solved using FLUENT and the particle motion is calculated using EDEM. The materials in the simulation are set according to the properties of the actual materials to ensure the authenticity of the simulation.

The periodic ripples for of erosion is explained by ripple formation model of Karimi A and Schmid R K. (Page 15)

Appendix D

Written Answer for the Comment 2

Dear Comment 2

Thank you for your letter. We thank you for the time and effort that you have put into reviewing the previous version of the manuscript. Your suggestions have enabled us to improve our work. Based on the instructions provided in your letter, we uploaded the file of the revised manuscript. Accordingly, we have uploaded a copy of the original manuscript with part of the changes highlighted in MS Word.

Appended to this letter is our point-by-point response to the comments. The comments are reproduced and our responses are given directly afterward in a different color. We hope that the revised manuscript is accepted for publication in the ROYAL SOCIETY OPEN SCIENCE.

Sincerely,

Yi Li, Hebing Zhang, Zhe Lin, Zhaohui He, Jialiang Xiang, Xianghui Su
linzhe0122@zstu.edu.cn

Reviewer comments 2

1. In equation 4.7, what was the value of K for the computation? Why?

Re: Thank you for your suggestion .

The wear constant is generally calculated through experimentation of the material, the wear constant ($k = 3.685e-4$) was calculated from the data of B. K. Prasad .

(*Prasad B K, Prasad S V, Das A A. Mechanisms of material removal and subsurface work hardening during low-stress abrasion of a squeeze-cast aluminium alloy-Al₂O₃ fibre composite[J]. Materials Science & Engineering A, 1992, 156(2):205-209.doi: 10.1016/0921-5093(92)90152-Q*)

Data of B. K. Prasad about abrasive wear rate of the base Al alloy.

Corresponding sliding distance(m)	Abrasion rate*10 ² (mm ³ /m)	Applied load (N)
392	25.5777712	22
784	20.27832978	
1176	16.77439734	
1568	14.37360805	
1960	14.86553325	
2352	13.42736276	
2744	13.5058117	

The specific calculation method is as follows :

Archard wear model:

$$W = K_s \frac{P}{P_m}$$

From this equation we can get the formula for K:

$$K = \frac{WP_m}{sP}$$

From the data of B. K. Prasad:

$$\frac{W}{s} = 1.351 * 10^{-4} \text{ mm}^3 / \text{mm}$$

$$P_m = 60 \text{ N} / \text{mm}^2$$

$$P = 22 \text{ N}$$

So the value of K is:

$$K = \frac{WP_m}{sP} = 3.685 \text{e-}4$$

2. The software versions of EDEM and FLUENT should be mentioned in the manuscript.

Re: Thank you very much for your comments. We have already indicated software versions of EDEM and FLUENT in the manuscript. The software versions of EDEM is 2.7, the software versions of FLUENT is 15.0. (Page 7)

3. The computational model was unsteady and the erosion tests were dependent upon time, so all the results should depend upon time from Fig. 10 to Fig. 14. For example in Fig. 10, it is impossible that the erosion rates keep constant all the time.

Re: Thank you for your suggestion.

In terms of experiment, each group of experiments is carried out five times and

the duration of a wear experiment is same. Then we calculate the average erosion rate for each group by mass difference before and after the experiment.

In terms of numerical simulation, all the research data was selected after the erosion was stabilized. This erosion stability means that the flow field has stabilized and the particles are filled with the entire bend. In our study, we did not put particles from 0s to 0.1s. During this period, the flow field calculation was stable. After 0.1s, the 'particle factory' began to generate a certain mass flow particles into the bend. From 0.2s to 1s, the particles have completely entered the elbow, and the erosion has been stabilized. At this time, the wall erosion is counted every 0.001s, and the average wear rate during this period is calculated.

In Fig. 10 and Fig. 12, the experimental and numerical simulation data are average erosion rates for a period of time. The numerical simulation data is from the beginning of wear stability.

In Fig. 11, the figure shows the position and motion of the particles at four mass concentrations at 1s results from CFD.

In Fig. 13, the figure shows the surface topography of the central position of the test sheet at different distances after 25 minutes of experimentation.

In Fig. 14, the figure shows the surface topography (after 25-minute experiment), wear cloud (after 1-s simulation) and particle position in the flow field (simulation at 1s) at 8 cm for mass concentrations.

Appendix E

Comments on Manuscript Number: RSOS-181254

Title: Relationship between Wear Formation and Large particle Motion in an Elbow

While the paper has improved, there are still a few points that need to be clarified. There are:

- On the numerical simulations part, the author noted that $Re=25770$ and the flow is in turbulent state of motion and the k-e model was used. They mention about two-way coupling in their response but there is no discussion in manuscript. There is no discussion on the interaction of turbulence with particles. The authors seem not to realize that the k-e model solves for the average velocity and the instantaneous turbulence fluctuation need to be accounted for. While for large particles of about 3mm used in this study the effect of turbulence may be small it need to be discussed.
- The discussions of flow and particle boundary conditions are improved. The authors stated that the standard wall function was used. But does standard wall function valid for a two-way coupled model?
- It is not quite clear what is the merit of showing 130 sample realizations in Figure 8.
- The experimentally observed ripple patterns are interesting. Karimi and Schmid [25] attributed these to cavitation and turbulent eddies. The authors' model does not include either but somehow shows the ripple effect. This need to be clarified.

Appendix F

Editor's Comments

Dear Dr. Li,

On behalf of the Editors, I am pleased to inform you that your Manuscript RSOS-181254.R1 entitled "Relationship between Wear Formation and Large-particle Motion in a Pipe Bend" has been accepted for publication in Royal Society Open Science subject to minor revision in accordance with the referee suggestions. Please find the referees' comments at the end of this email.

The reviewers and Subject Editor have recommended publication, but also suggest some minor revisions to your manuscript. Therefore, I invite you to respond to the comments and revise your manuscript.

Associate Editor Comments

Please address all of the remaining points identified by the reviewers in a further revision of your paper.

Responses to Reviewers

Dear Editor and Reviewers,

Thank you for your effort in processing and reviewing the manuscript. Based on the comments from the reviewers, we have revised our manuscript to our best. The revised manuscript is uploaded for your further consideration, and the letter of response to reviewers is also provided in a point-to-point form.

We would like to thank you for allowing us to resubmit a revised copy of the manuscript. We hope that the revised manuscript is acceptable for publication in the ROYAL SOCIETY OPEN SCIENCE.

Sincerely,

Yi Li, Heping Zhang, Zhe Lin, Zhaohui He, Jialiang Xiang, Xianghui Su
linzhe0122@zstu.edu.cn

Reviewer comments 1

While the paper has improved, there are still a few points that need to be clarified. There are:

1. On the numerical simulations part, the author noted that $Re=25770$ and the flow is in turbulent state of motion and the k-e model was used. They mention about

two-way coupling in their response but there is no discussion in manuscript. There is no discussion on the interaction of turbulence with particles. The authors seem not to realize that the k-e model solves for the average velocity and the instantaneous turbulence fluctuation need to be accounted for. While for large particles of about 3mm used in this study the effect of turbulence may be small it need to be discussed.

Re: Thank you for your constructive comment. We have added the governing equations of the fluid and equations of forces from particles to fluid according to the comment and have highlighted the revised part in yellow text (Page5, Page6, Page7). Because of the particle size (3mm) is large in this research, the inertial force dominates the particle motion rather than the instantaneous turbulence pulsation. So the k-e model was used in the numerical simulation of flow field to solve the average velocity.

2. The discussions of flow and particle boundary conditions are improved. The authors stated that the standard wall function was used. But does standard wall function valid for a two-way coupled model?

Re: Two calculations of flow field and particle motion are not simultaneous (shown in figure 4). The standard wall function is used to calculate the flow field of the near-wall region. After one time step of flow field calculation, the calculated data is transferred to calculate the particle motion. In this calculation stage, the standard wall function does not appear. So it does not affect the two-way coupling calculation process.

3. It is not quite clear what is the merit of showing 130 sample realizations in Figure 8

Re: We are grateful for the suggestion. We have added explanation of 130 samples according to the comment and have highlighted the revised part in yellow text (Page 10). The average velocity of 130 particles is calculated to describe the motion rule of the particles in the curved region. And 130 samples are selected to eliminate accidental errors.

4. The experimentally observed ripple patterns are interesting. Karimi and Schmid [25] attributed these to cavitation and turbulent eddies. The authors' model does not include either but somehow shows the ripple effect. This need to be clarified.

Re: Thank you for your comment. We refer to the explanation of Karimi and Schmid about ripple generation. In the study of Karimi and Schmid, the turbulent eddy near the wall dominates the motion of the particles (120 microns). But due to the particle size (3mm) is relatively large in this study, the vortex near the wall does not dominate the particle motion. In addition, the inertia has an important effect on particle motion. And the velocity difference between the near-wall fluid and the bulk fluid will bring the deviation of particle motion trajectory. The

combination effect of these factors causes particles to bounce forward on the wall surface regularly (shown in figure 15(a)). Over a period of time, ripples appear.

Reviewer comments 2:

1. It is suggested to simply add the explanation of average erosion rate for each group by mass difference before and after the experiment.

Re: Thank you for your constructive comment. In the Dryad database, we have carefully added data from five sets of experiments for each concentration, including maximum erosion and average erosion. In addition, we have added some explanation of the wear results obtained from the experiment and have highlighted it in yellow text.

2. It is also suggested to simply add the explanation of all the computational erosion data was selected after the erosion was stabilized.

Re: Thank you for your comment. We have added it according to the comment.

(Page 12)